

PeerJ Hubs
Published on behalf of

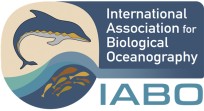

International Association for Biological Oceanography
IABO

# Cleaning symbiosis in coral reefs of Jardines de la Reina National Park

Andy Joel Corso[1,*], Fabián Pina-Amargós[2] and Leandro Rodriguez-Viera[3,*]

[1] Center for Marine Research, University of Havana, Havana, Cuba
[2] Blue Sanctuary-Avalon, Ciego de Ávila, Cuba
[3] Faculty of Marine and Environmental Sciences, Campus de Excelencia Internacional del Mar (CEIMAR), University of Cadiz, Puerto Real, Cádiz, Spain
[*] These authors contributed equally to this work.

## ABSTRACT

**Background.** Cleaning symbiotic interactions are an important component of coral reef biodiversity and the study of the characteristics of these interacting species networks allows to assess the health of communities. The coral reefs of Jardines de la Reina National Park (JRNP) are subject to a protection gradient and there is a lack of knowledge about the effect of different levels of protection on the cleaning mutualistic networks in the area. The present study aims to characterize the mutualistic cleaning networks in the reefs of JRNP and to assess the potential effect of the protection gradient on their characteristics.

**Methods.** We visited 26 reef sites distributed along the National Park and performed 96 band transects (50 m × 4 m). Low, medium and highly protected regions were compared according to the number of cleaning stations and the abundance and number of species of clients and cleaners associated with them. Additionally, we built interaction networks for the three regions and the entire archipelago based on a total of 150 minutes' video records of active cleaning stations. We assessed ecological networks characteristics (specialization, nestedness) using network topological metrics.

**Results.** We found a high diversity and complex cleaning interaction network with 6 cleaner species and 39 client species, among them, the threatened grouper *Epinephelus striatus* was one of the most common clients. No clear effect of the protection level on the density, abundance or diversity of cleaners and clients was detected during this study. However, we found that the network structure varied among regions, with the highly protected region being more specialized and less nested than the other regions. Our research reveals some patterns that suggest the effect of fishing pressure on cleaning symbiosis, as fishing may reduce the abundance and composition of client species, especially those that are targeted by fishers. However, fishing pressure may not be the main factor influencing cleaning symbiosis inside of the National Park, as other factors, such as habitat quality or environmental conditions may have stronger effects on the demand for cleaning services and the interactions between cleaners and clients. Our research provides insights into the factors that influence cleaning symbiosis and its implications for coral reef conservation and management.

Corresponding author
Leandro Rodriguez-Viera,
leokarma@gmail.com

## INTRODUCTION

Mutualistic symbioses are essential relationships in biological communities. They are closely related to trophic cascades and influence growth, adaptability of individuals, diversity and abundance (*Schleuning, Fründ & García, 2015*). Cleaning mutualisms are defined as those interactions in which one organism, termed "cleaner", provides another, a "client", with a cleaning service by feeding on its parasites and significantly reducing its parasitic load (*Grutter, 1995*). Ectoparasites negatively affect fish hosts by altering size, growth, larval recruitment success, and survival (*Smit, Bruce & Hadfield, 2019*). Thus, the presence of cleaners in reefs implies remarkable benefits to the health of communities (*Bshary et al., 2003*; *Grutter & Bshary, 2003*; *Clague et al., 2011*). Although cleaning symbioses in coral reefs have been studied, only a few investigations have employed more holistic approaches that allow the analysis of cleaning mutualistic interaction networks on reefs as complex systems possessing emergent properties with their own study methods (*e.g., Guimarães et al., 2006*; *Sazima, Grossman & Sazima, 2010*; *Quimbayo et al., 2017*; *Quimbayo et al., 2018a*; *Quimbayo et al., 2018b*).

Mutualistic ecological networks represent the interactions between two functional groups, such as plants and their floral visitors or as cleaners and clients (*Bascompte, 2009*). By studying these ecological networks, it is possible to quantify and describe how individuals of different species interact, estimate the diversity of interactions, determine the structural organization of the mutualistic network or analyze the degree of trophic specialization (*Luna et al., 2020*). The descriptors used to characterize networks are useful tools that facilitate the analysis of the temporal and spatial variation of species and their interactions (*Dormann, Fründ & Schaefer, 2017*). Thereby, descriptors allow the comparison of interactions of the same type in different habitats by describing variations in diversity, structural organization, and specialization (*Tylianakis & Morris, 2017*).

The Caribbean Sea is considered one of the most biodiverse regions on the planet (*Roberts et al., 2002*; *Miloslavich et al., 2010*). However, Caribbean countries are among the most exposed to the effects of climate change, and therefore the conservation of various habitats and species that are already showing signs of deterioration is also at risk (*Caballero-Aragón et al., 2019*). Although Cuba does not escape this scenario, biodiversity conservation in the archipelago is certainly superior, which is a consequence of the implementation of successful environmental policies and the existence of an extensive network of protected areas (*Galford et al., 2018*). Jardines de la Reina National Park, with an area of approximately 950 km$^2$ and a high degree of protection, stands out as one of the best-conserved areas in the Caribbean (*Jackson et al., 2014*; *Pina-Amargós et al., 2014*). Its coral reefs and reef fish communities are in good condition, despite years of sustained fishing pressure in the area (*Navarro-Martínez & Angulo-Valdés, 2015*).

However, *Pina-Amargós et al. (2014)* suggested the existence of different degrees of protection effectiveness in the MPA based on the spatial pattern of abundances of different fish species. The authors attribute this effect to the incidence of illegal fishing pressure at the eastern and western boundaries of the reserve and the greater effectiveness of protection in the central area (*Pina-Amargós, González-Sansón & Cabrera-Páez, 2008*; *Pina-Amargós et*

*al., 2014*; *Hernández-Fernández et al., 2019*). Thus, the existence of a gradient of protection on the fish community in Jardines de la Reina National Park offers the opportunity to verify its effect on the characteristics of mutualistic interaction networks at various sites that differ in the availability of potential clients. Likewise, although cleaning mutualisms have been extensively studied in the Caribbean Sea (*Vaughan et al., 2017*), no published research includes the Cuban archipelago as a study area. Thus, the present research aims to answer the following questions: what are the characteristics of the interaction networks of cleaning mutualisms in Jardines de la Reina National Park coral reefs? What is the effect of MPA's different levels of protection effectiveness on coral reef cleaning mutualisms?

## MATERIALS & METHODS

### Study area

The Jardines de la Reina archipelago (Fig. 1) currently has more than 950 km$^2$ under the National System of Protected Areas in Cuba, with the management category of National Park (*Pina-Amargós et al., 2014*) and consists of an extensive chain of cays associated with mangrove islands and stretches of coral reef facing the Caribbean Sea. The reef structure in this area includes a reef terrace between 5 and 15 m deep, followed by a rocky slope culminating in a sandy esplanade, with the formation of large spur and groove area and a reef drop-off (*Navarro-Martínez et al., 2022*).

The sampling effort was concentrated on the largest group of cays in the archipelago, *Las Doce Leguas*, during March 2019, December 2021, and August 2022. A total of 26 sites continuously distributed from the far-East to the far-West of the MPA were sampled. For analysis purposes, sites were grouped into three main regions according to different levels of protection effectiveness: low protection level (eastern region), medium protection level (western region) and high protection level (central region) (Fig. 1, Table 1). This artificial grouping is similar to that applied by *Pina-Amargós, González-Sansón & Cabrera-Páez (2008)* and *Pina-Amargós et al. (2014)* and facilitates the evaluation and comparison of the results. Studied reefs are located south of the line of cays and include two ecosystems: the crest reef and the reef cliff. The sampling depth did not exceed 25 m.

### Sampling methodology

Between one and four band transects of 50 m × 4 m were conducted at each site, representing a total of 19,200 m$^2$ monitored ($n = 96$). In each transect, the number of cleaning stations, the cleaner species and clients associated, and the number of individuals of each species was quantified.

Cleaning stations were considered as such when the reef sites that met the typical characteristics and conditions (*i.e.* coral, sponge or rock promontories, crevices, large anemones) and where occurred the observation of pairs of client and cleaner species interacting or only by the presence of species of well-known role as dedicated cleaners (*Limbaugh, 1961*). The client and cleaner diversity in each transect is the number of species of each role detected on transects associated with cleaning stations. The total abundance of cleaners and clients is the number of individuals, regardless of species, detected acting these roles associated with cleaning stations.

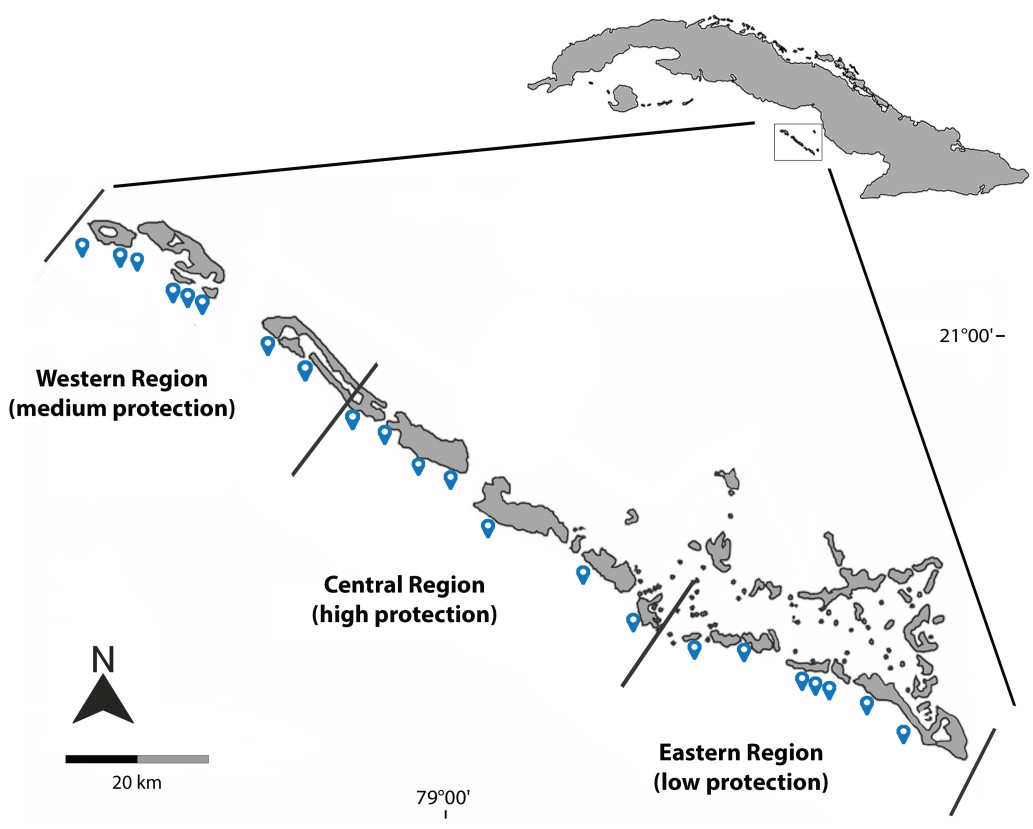

**Figure 1** **Geographical location of the sites sampled during March 2019, December 2021 and August 2022 in Jardines de la Reina National Park.** Blue dots represent the sampled sites; black lines delimit the three regions and different protection levels.

A team of two SCUBA divers conducted cleaning station surveys per transect. Measurements were started approximately five minutes after defining the transect positioning to minimize behavioral disruptions. Cleaning stations and individuals were counted as long as they were observed in front and to the sides of the observer, and most species were identified *in situ*.

Video recordings were made to identify the species involved and the number of times each pair of species interacted. Between 40 and 60 min of video recording was collected for each region. According to the criteria of *Losey (1972)* and *Quimbayo et al. (2018b)*, we considered that there were cleaning interactions when physical contact between client and cleaner was detected. Client species were classified in functional groups following the criteria of *Micheli et al. (2014)* and *Navarro-Martínez et al. (2022)* with modifications according to *Choat, Clements & Robbins (2004)*, *Bonaldo, Hoey & Bellwood (2014)* and *Adam et al. (2015)* and threat category according to the International Union for Conservation of Nature (IUCN). Functional groups were based on the combination of two functional traits: mobility and feeding guild as previously described in *Navarro-Martínez et al. (2022)* (Table S1). Specifically, the mobility categories were pelagic roving, cryptic, and midwater. Feeding guilds were based on nine categories: macroinvertivores (invertebrate feeders of

**Table 1** Sampled sites in Jardines de la Reina National Park between 2019 and 2022, and the number of transects carried out per site.

| MPA region | Site | Longitude | Latitude | Number of transects per site |
|---|---|---|---|---|
| **East** | La Mexicana | −78°45′ | 20°58′ | 1 |
| (lower protection) | Macao | −78°40′ | 20°53′ | 2 |
| | Peralta | −78°51′ | 20°60′ | 2 |
| | Boca Rica | −78°48′ | 20°59′ | 2 |
| | Boca Seca | −78°51′ | 20°60′ | 1 |
| | Los Hierros | −78°67′ | 20°64′ | 5 |
| | Carabinero | −78°61′ | 20°63′ | 1 |
| **Center** | Cachiboca | −78°75′ | 20°67′ | 2 |
| (higher protection) | Anclitas | −78°91′ | 20°77′ | 5 |
| | Las Cruces | −78°81′ | 20°72′ | 2 |
| | Mogotes | −78°80′ | 20°72′ | 4 |
| | Farallón | – | – | 4 |
| | Intermedio | – | – | 4 |
| | Mariflores | −78°81′ | 20°72′ | 4 |
| | Las Auras | −79°06′ | 20°87′ | 2 |
| | Bayameses | −79°10′ | 20°90′ | 7 |
| | Peruano | −79°02′ | 20°84′ | 2 |
| | Los Pinos | −78°99′ | 20°83′ | 2 |
| **West** | Boca Grande | −79°22′ | 20°97′ | 5 |
| (medium protection) | Boca de Guano | −79°16′ | 20°94′ | 3 |
| | Cinco Balas | −79°31′ | 21°03′ | 5 |
| | Casimba | −79°31′ | 21°03′ | 6 |
| | Alcatraz | −79°30′ | 21°02′ | 7 |
| | Bretón | −79°.48′ | 21°08′ | 6 |
| | Esterón | −79°.40′ | 21°07′ | 6 |
| | Horqueta | −79°.39′ | 21°07′ | 6 |
| **Total sampling units** | | | | 96 |

25–50 cm maximum length), microinvertivores (small invertebrate feeders of < 30 cm length), sessile invertivores (sessile invertebrate feeders), predators (invertebrate feeders and piscivores), piscivores, omnivores, planktivores, browsers, and grazers.

## Statistical analysis

To assess the existence of a possible effect of the protection gradient on symbiotic cleaning interactions, MPA regions were compared according to number of cleaning stations, diversity of cleaner species, diversity of client species, total abundance of cleaners and total abundance of clients. This was done by using a smaller sample size ($N = 37$) than the initial one, so that three regions had an approximately similar number of sample units. Variables were tested for homoscedasticity (Levene test) and to verify if they fitted to a normal distribution (Kolmogorov–Smirnov test). Non-parametric Kuskal-Wallis test was used to verify statistical differences ($\alpha = 0.05$) among regions.

Mutualistic interaction matrices between client species and cleaners were generated for each sector and the whole archipelago. Bipartite interaction networks were constructed for each region and for the MPA. The networks were generated from adjacency matrices with values for the number of interactions between pairs of species. The networks, due to their bipartite structure, consist of two groups of species segregated in different columns. Each column is formed by nodes, representing the species, the link between nodes represents the interaction and the thickness of the link is proportional to the number of interactions.

We calculated connectance, nestedness, $H_2'$ specialization (*Blüthgen, Menzel & Bluthgen, 2006*), niche overlap and complementarity for all networks. Connectance is calculated as proportion of realized interactions given a total number of possible links (*Quimbayo et al., 2018a*). Nestedness was assessed using *NODF* metric and it describes asymmetrical networks with some species cleaning the most of clients whereas others clean just a subset of those clients (*Guimarães et al., 2006*). $H_2'$ measures network-level specialization ranging between 0 (no specialization) and 1 (perfect specialization (*Dormann et al., 2009*). Niche overlap indicates similarities in interaction pattern between species from the same trophic levels whereas functional complementarity measures the ecological niche complementarity between species of the same trophic level as the total branch length of a dendrogram based on differences in interacting species assemblages (*Devoto et al., 2012*). These indices are generally used to describe the characteristics and structure and infer the robustness and the specialization of interaction networks (*Quimbayo et al., 2018a*). Processing was performed using R programming environment and bipartite package (*R Core Team, 2023*; *Dormann, Gruber & Fründ, 2008*).

## RESULTS

### Interaction ecology

During our surveys, we recorded 45 species engaging in cleaning mutualism interactions on the reefs of the Jardines de la Reina National Park. Of these, 39 species were identified as clients, including the taxonomic entities *Pterois* spp. and *Kyphosus* spp. whose *in situ* identification to species level proved difficult. Client species belong to 15 families (Table S1) and those with higher number of species were Labridae (eight species), Serranidae (seven species), and Pomacentridae (four species). Six species belonging to two families of fish: labridae (two species), and Gobiidae (two species); and two families of shrimps: Palaemonidae (one species), and Stenopodidae (one species) were identified as cleaners. *In situ* identification of *Elacatinus genie* and *E. evelynae* without disturbing cleaning behaviors was difficult, so they were treated as the *Elacatinus* spp. (Table 2).

During the study, 69 different combinations of cleaner –client pair species were detected. Cleaner fish, *Bodianus rufus* (Labridae) interacted with a total of 23 different client species, the *Elacatinus spp.* with 20 species, *Thalassoma bifasciatum* with 17, and the shrimps *Stenopus hispidus* and *Ancylomenes pedersoni* with six and three species, respectively. In the case of the clients, the species that interacted with the highest number of cleaner species were the serranids *Epinephelus striatus* and *Mycteroperca tigris* with five and four species respectively.

**Table 2** List of cleaner species detected on coral reefs in Jardines de la Reina National Park, Cuba, from surveys conducted between 2019 and 2022. The studies where they were first reported expressing cleaning behavior are presented.

| Species | Report | Cleaning lifestyle |
|---|---|---|
| **Gobiidae** | | |
| *Elacatinus evelynae* (Böhlke & Robins, 1968) | *Whiteman & Côté (2002)* | Dedicated |
| *Elacatinus genie* (Böhlke & Robins, 1968) | *Colin (1975)* | Dedicated |
| **Labridae** | | |
| *Thalassoma bifasciatum* (Bloch, 1791) | *Eibi-Eibesfeldt (1955)* | Facultative |
| *Bodianus rufus* (Linneaus, 1758) | *Eibi-Eibesfeldt (1955)* and *Limbaugh (1961)* | Facultative |
| **Palaemonidae** | | |
| *Ancylomenes pedersoni* (Chace, 1958) | *Limbaugh, Pederson & Chace Jr (1961)* | Dedicated |
| **Stenopodidae** | | |
| *Stenopus hispidus* (Olivier, 1811) | *Jonasson (1987)* | Dedicated |

In terms of number of interactions, a total of 588 interactions were recorded. The most frequent cleaner species was *B. rufus* (Labridae) with 39% of interactions, followed by juveniles of *T. bifasciatum* (Labridae) with 35% of interactions, *Elacatinus* (Gobiidae) with 21% and the decapods *A. pedersoni* (Palaemonidae) and *S. hispidus* (Stenopodidae) whose interactions together represent 2% of the total (Fig. 2). According to *Vaughan et al. (2017)* three of the detected species are considered dedicated cleaners; these are *A. pedersoni*, *Elacatinus* spp., while the remaining cleaners (*T. bifasciatum*, *B. rufus*, and *S. hispidus*) express this behavior facultatively (Table 2).

The most frequently detected client species establishing interactions were *Clepticus parrae* (Labridae) with 30% of the total, followed by *Caranx ruber* (Carangidae) with 12%, and *Acanthurus coeruleus* (Acanthuridae), and *Epinephelus striatus* (Serranidae) both with 7%. These species constituted more than 50% of the total interactions detected in the study. *C. parrae* and *T. bifasciatum*, with 21% of the total interactions, constituted the cleaner-client pair of species that interacted the most; followed by *C. ruber* and *B. rufus* whose interactions accounted for 7% of the total. These were followed by the pairs *C. parrae* and *B. rufus*, *A. coeruleus* and *B. rufus*, and *E. striatus* and *Elacatinus* spp. with 7%, 5%, and 4% of the total interactions respectively.

We identified 15 different client functional groups. Roving macroinvertivores presented the highest number of species associated with cleaning stations (with seven species), followed by roving grazers (six) and roving piscivores (five) (Table S1). Regarding the frequency of interactions established by each group, the highest percentage of interactions was presented by the group of midwater planktivores with 30% of the total interactions, followed by roving herbivores with 25% and then, midwater piscivores with 11.5%.

Analysis of the number of interactions of cleaner species with the functional groups of clients revealed that *B. rufus* interacted with clients from 15 functional groups, and more than 50% of its interactions were established with roving grazers and midwater piscivores and planktivores. *T. bifasciatum* interacted with species from 11 functional groups; however,

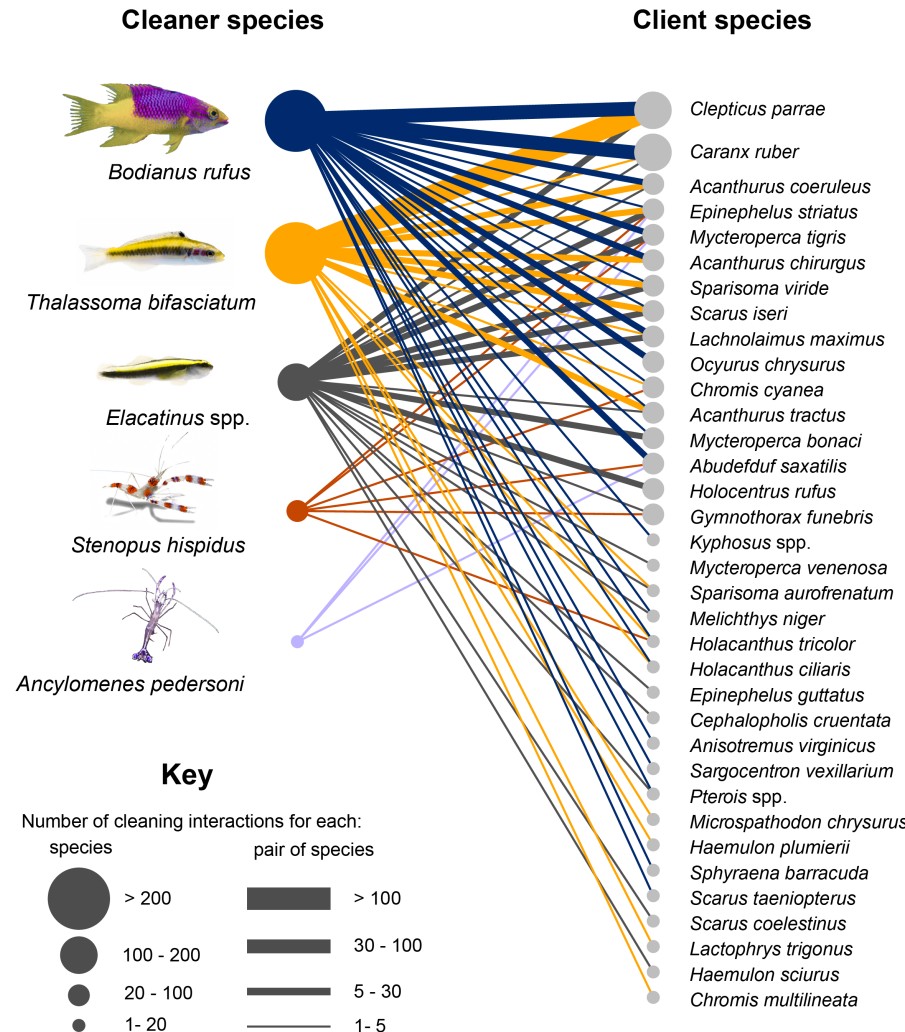

**Cleaner species**

*Bodianus rufus*

*Thalassoma bifasciatum*

*Elacatinus* spp.

*Stenopus hispidus*

*Ancylomenes pedersoni*

**Client species**

*Clepticus parrae*
*Caranx ruber*
*Acanthurus coeruleus*
*Epinephelus striatus*
*Mycteroperca tigris*
*Acanthurus chirurgus*
*Sparisoma viride*
*Scarus iseri*
*Lachnolaimus maximus*
*Ocyurus chrysurus*
*Chromis cyanea*
*Acanthurus tractus*
*Mycteroperca bonaci*
*Abudefduf saxatilis*
*Holocentrus rufus*
*Gymnothorax funebris*
*Kyphosus* spp.
*Mycteroperca venenosa*
*Sparisoma aurofrenatum*
*Melichthys niger*
*Holacanthus tricolor*
*Holacanthus ciliaris*
*Epinephelus guttatus*
*Cephalopholis cruentata*
*Anisotremus virginicus*
*Sargocentron vexillarium*
*Pterois* spp.
*Microspathodon chrysurus*
*Haemulon plumierii*
*Sphyraena barracuda*
*Scarus taeniopterus*
*Scarus coelestinus*
*Lactophrys trigonus*
*Haemulon sciurus*
*Chromis multilineata*

**Key**

Number of cleaning interactions for each:

| species | pair of species |
|---|---|
| > 200 | > 100 |
| 100 - 200 | 30 - 100 |
| 20 - 100 | 5 - 30 |
| 1- 20 | 1- 5 |

**Figure 2** **Cleaning interaction network in Jardines de la Reina National Park.** Circles are directly proportional to the number of interactions established by species. Links between nodes indicate existence of interactions, width of the link are directly proportional to the number of interactions established between pairs of species.

almost 90% of the time it cleaned midwater planktivores and roving grazers. The *Elacatinus* spp. clients belonged to 10 different functional groups and, although the number of interactions is more homogeneously distributed than in the rest of the species, in about 90% of the observations cleaning gobies interacted with piscivores, predators and roving grazers. Shrimp species *S. hispidus* and *A. pedersoni* interacted with species from seven and three functional groups, respectively. In both cases, roving piscivores predominated.

## Effect of protection gradient on cleaning symbiosis

A total of 230 cleaning stations ($N = 96$) were observed throughout the study, with a mean density value of $2.3 \pm 0.5$ stations/200 m². This variable presented a similar pattern throughout the entire area of the PNJR and did not show statistically significant differences
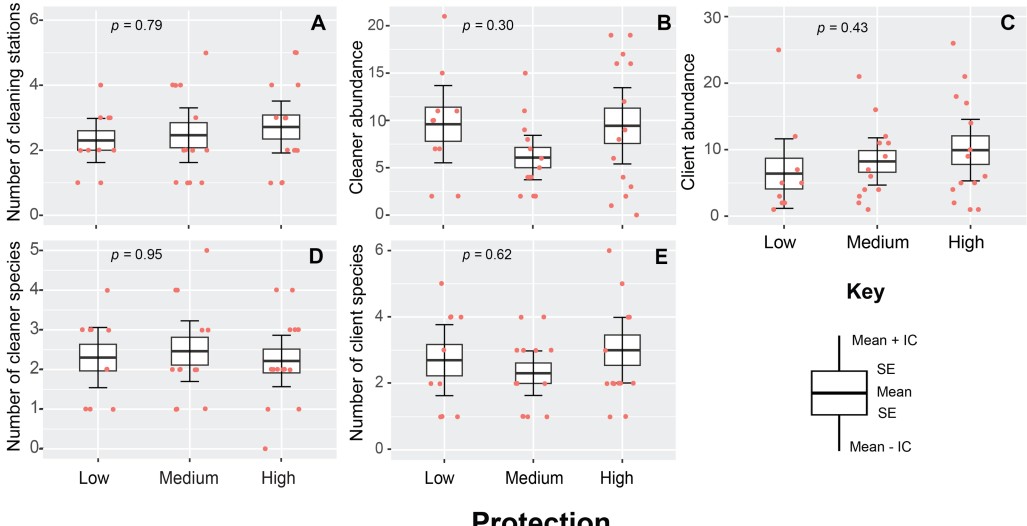

**Figure 3 Cleaning stations variables measured in Jardines de la Reina National Park during sampling in 2019.** (A) Number of cleaning stations. (B) Cleaner abundance. (C) Client abundance. (D) Number of cleaner species. (E) Number of client species. The boxes represent the interquartile range, the black line represents the median, and the vertical line represents the median ± 1.5 times the interquartile range. The raw data points and *p*-value from generalized linear model fitting are also shown.

among MPA regions ($p = 0.79$, $H = 0.46$, Fig. 3A). However, highly protected region exhibit the highest medium density value of cleaning stations with 2.7 ± 1.3 stations/200 m², whereas east and west regions show similar densities of stations.

In the case of cleaner abundances, the differences found among regions were not statistically significant ($p = 0.31$, $H = 2.43$, Fig. 3B). Medium protection region exhibited the lowest mean abundances, whereas high and low protection region presented very similar values of abundances with 9.6 ± 1.7 and 9.5 ± 1.6 individuals respectively. Client abundance was also statistically similar across three regions of MPA ($p = 0.43$, $H = 1.67$, Fig. 3C). However, this variable presented the same pattern observed in number of cleaning stations, high protection region exhibited the highest mean abundances with 10 ± 2.4 individual clients, whereas the lowest mean client abundance values was registered in the lowly protected region with 6.5 ± 2 individuals.

The diversity of cleaners ($p = 0.95$, $H = 0.10$, Fig. 3D) and clients ($p = 0.62$, $H = 0.95$, Fig. 3E) was also not affected by the level of protection. However, we noticed that the high protection region had the highest mean values for client species number, with a mean of 3 ± 1.5 species in each transect and records of up to 6 client species.

The networks differed in their degree of specialization and nestedness according to the level of protection of the regions. The network of the high protection region had the highest H2′value (0.55) and the lowest connectance value (0.33), indicating that it was the most specialized network. The network of the low protection region had the lowest H2′value (0.41) and the highest connectance value (0.53), indicating that it was the least specialized network. The JRNP network had intermediate values for both metrics (0.39 and 0.54,
**Table 3** Descriptor indices of cleaning interaction networks generated from analysis of video recording cleaning stations on coral reefs in the Jardines de la Reina National Park, Cuba.

| Network | Conectance | NODF (Nestedness) | H2′ specialization | Niche overlap | Functional complementarity |
|---|---|---|---|---|---|
| East region (lower protection) | 0.533 | 51.7 | 0.413 | 0.453 | 95 |
| Central region (higher protection) | 0.333 | 50.5 | 0.560 | 0.238 | 262 |
| West region (medium protection) | 0.341 | 50.2 | 0.502 | 0.315 | 144 |
| JRNP | 0.394 | 54.9 | 0.411 | 0.316 | 486 |

respectively). All networks had similar nestedness values, with the JRNP network having the highest NODF value (54.9) (Table 3). Functional redundancy and niche overlap are measures of how similar the species are in their roles and interactions within the network. Functional redundancy reflects the extent to which species can be substituted by others without affecting the network structure or function, while niche overlap measures the degree to which species share the same resources or partners in the network. The network of the high protection region had the lowest niche overlap value (0.23) and the highest functional complementarity value (262.4), suggesting that its species were more distinct and less replaceable than those of the other networks.

# DISCUSSION

## Interaction ecology

The 45 species of fish and crustaceans recorded in this research have been previously reported for the JRNP and fish represent 17% of the total species (*Pina-Amargós, González-Sansón & Cabrera-Páez, 2008*). Cleaner species detected have been previously reported to occupy that role (*Limbaugh, 1961*) (Table 3). Most studies in the Caribbean have noted the importance of *Elacatinus* and juvenile individuals of *T. bifasciatum* in cleaning stations (*Longley & Hildebrand, 1941*; *Losey, 1972*; *Darcy, Maisel & Ogden, 1974*; *Dunkley, Cable & Perkins, 2018*). However, in this study *B. rufus* and *T. bifasciatum* were the main cleaners.

According to *Guimarães et al. (2006)*, the ratio between species richness of clients and cleaners is the typically observed in reef cleaning networks. The total number of species involved in the interactions (45) indicates that PNJR cleaning mutualism network is a species-rich network, whose number of species in both groups exceeds those described in others in the region with similar sampling efforts (*Johnson & Ruben, 1988*). However, both cleaner and client richness could increase since that studies in Caribbean region (*Titus et al., 2019*; *Dunkley et al., 2019*) report interactions of cleaners detected in this research with client species inhabiting PNJR reefs that were not observed in cleaning stations. For example, species such as *Pomacanthus paru*, present in the MPA reefs (*Pina-Amargós, González-Sansón & Cabrera-Páez, 2008*) and frequently reported as cleaner in their juvenile phase (*Vaughan et al., 2017*), were not observed expressing cleaning behaviors during the study, which indicates more sampling effort is required.

## Effect of protection gradient on cleaning symbiosis

The absence of significant differences among regions in the variables related to cleaning symbiosis may be explained by the fact that the protection gradient does not have a strong impact on the fish species that participate in this mutualistic interaction. These results are similar to those reported by *Silvano, Tibbetts & Grutter (2012)*, who found that cleaner fish density, as well as the number of individuals and species cleaned, did not differ significantly bet ween a fished and a no-take site on the Great Barrier Reef. Therefore, the abundance and diversity of cleaners and clients may be more influenced by other factors that are independent of the level of protection, such as habitat complexity, benthic community composition, ectoparasite infection rates, and interspecific interactions.

Habitat complexity is one of the main factors that determine the number and distribution of cleaning stations, as they provide suitable substrates and shelters for cleaners and clients (*Whittney et al., 2021*). The coral reefs sites of JRNP have similar habitat complexity and benthic community composition throughout the park, as reported by *Pina-Amargós, González-Sansón & Cabrera-Páez (2008)* and *Hernández-Fernández et al. (2019)*. This may explain why the density of cleaning stations did not vary significantly among regions. However, the high protection region had a slightly higher density of cleaning stations, which may be related to the higher abundance of fish in this region (*Navarro-Martínez et al., 2022*). Higher fish abundance may imply higher ectoparasite infection rates, which increase the demand for cleaning services and the availability of food resources for cleaners (*Marcogliese, 2002*; *Sikkel, Cheney & Côté, 2004*). This may enhance the trophic niche and abundance of cleaners, especially those that are dedicated cleaners and prefer ectoparasites in their diet, such as *Elacatinus* cleaning gobies (*Soares et al., 2008*; *Soares et al., 2010*).

We also expected to find higher abundance and species richness of clients in the highly protected region. However, we did not observe significant differences in these variables across the protection gradient. One possible explanation is that the competitive interactions among clients and cleaners at the cleaning stations may have masked the effect of protection. Cleaning stations are not homogeneous in size and structure, but they depend on the species of cleaners that inhabit them (*Huebner & Chadwick, 2012*; *Whittney et al., 2021*). However, this variation does not seem to influence the number of cleaners (*Whiteman & Côté, 2004*) or clients that use the stations at any given time (either as cleaners or as recipients of cleaning services). Therefore, the number of clients at the stations may be regulated by the availability of fish from the surrounding community that may seek cleaning services. This is consistent with the finding that the abundance of clients and cleaners is the main predictor of the visitation rate to the cleaning stations (*Floeter, Vázquez & Grutter, 2007*). Another factor that may limit the number of clients at the stations is the presence of predators or competitors that may deter potential clients from approaching the cleaners. This is supported by our results that showed that piscivorous and predatory species were among the most frequent visitors to the cleaning stations.

Another potential reason why we did not find a clear effect of protection on the variables related to cleaning symbioses is that protection may only affect some specific species that are targeted by fishing activities (*Pina-Amargós et al., 2014*). These species may not be involved in the cleaning mutualism network, or may have a minor role in it. Furthermore, the

cleaner-client networks are not tightly structured or obligatory (*Quimbayo et al., 2018a*), so the dependence between the two groups is low. Therefore, protection may not have a strong impact on the dynamics of cleaning symbioses in coral reefs. However, we must keep in mind that this work evaluated different degrees of protection in a marine protected area, where even the least protected area also has protection. It would be essential to evaluate and compare with marine areas without any protection and with a marked effect of fishing (*Silvano, Tibbetts & Grutter, 2012*).

## Network indices

Indices calculated for the networks in each region describe them in two important ways: structure and functional redundancy. In the first group, connectance provides information on how connected the two trophic levels that make up the network are in relation to the maximum value of possible connections between the nodes of both levels (*Dormann et al., 2009*). Higher connectance values mean that the network is more complete. We found that the network in the low protection region had the highest connectance, even though we expected the opposite. This may be related to the species richness in each region. More species means more nodes in the network, and therefore less chance of observing all the potential interactions. Since we recorded more species in the high protection region (23 species), we would expect that region to have lower connectance than the others. Nestedness shows how similar the patterns of interactions are among different subsets of species in the network (*Bascompte & Jordano, 2013*). Higher nestedness values indicate that there is a core group of species at each level that interact frequently with each other (generalists), and another group of species that interact less often and only with the core group (specialists). This way, the generalists provide a stable source of resources, in this case, cleaning services, that allows for specialization. Our results for nestedness are consistent with those reported for other cleaning networks in the Caribbean (*Quimbayo et al., 2018a*).

We also calculated three indices that reflect the functional redundancy of the network: H2′specialization, niche overlap and functional complementarity. These indices can be analyzed together, as they all indicate how similar or different the species are in their interactions with each other. In this case, the values we obtained for the archipelago network suggest that, as usual for these kinds of networks, there is a high degree of generalization and functional redundancy, which agrees with the findings of *Quimbayo et al. (2018a)*. These authors argue that cleaning mutualism networks exhibit a high degree of niche overlap and generalization, due to the non-specific nature of the interactions. In the JRNP mutualisms, this is due to the predominance of interactions established by the facultative cleaners *T. bifasciatum* and *B. rufus*. These species exploit other food sources and often clean up clients that are common or use the same habitat. In localities with only facultative cleaners, client use overlaps more because they clean when they are juveniles or when predation risk is low (*Côté, 2000*; *Vaughan et al., 2017*) and therefore eventually and sporadically interact with the entire client group.

Our results also suggest that the network in the high protection region is slightly less generalized and redundant, and more specialized and complementary. This may be related to the higher species richness and abundance of clients, which may create better conditions

for cleaners and clients to show more natural and specific behaviors and preferences related to cleaning symbiosis that are not possible in environments with fewer clients. Moreover, *Silvano, Tibbetts & Grutter (2012)* showed that the community of clients changed from a no-take site to a protected area. Therefore, cleaning symbiotic network structure may also vary according to the level of protection, as different client species may form different types and frequencies of links with cleaners.

## CONCLUSIONS

This research studied the network of cleaning interactions in the JRNP, a marine protected area in Cuba, and how it was affected by the protection level of different regions. We found a high diversity and complexity of cleaning symbiosis in the JRNP, but no clear effect of protection level on the cleaning stations density, abundance or diversity of cleaners and clients. However, we found that the network structure varied among regions, with the high protection region being more specialized and less nested than the other regions. Our research reveals some patterns that suggest the effect of fishing pressure on cleaning symbiosis, as fishing may reduce the abundance and composition of client species, especially those that are targeted by fishers. However, fishing pressure may not be the main factor influencing cleaning symbiosis inside of the National Park, as other factors, such as habitat complexity and environmental conditions, may have stronger effects on the demand for cleaning services and the interactions between cleaners and clients. Our research provides insights into the factors that influence cleaning symbiosis and its implications for coral reef conservation and management.

## ACKNOWLEDGEMENTS

The authors express their gratitude to the International Chair of Conservation and Management of Marine-Coastal Ecosystems of the Harte Research Institute of the University of Corpus Christi Texas A & M The Ocean Foundation, and Sweet-Avalon. We are also grateful to the crew of "MV Ocean for Youth" and the staff of Center for Marine Research at University of Havana for their assistance. Thanks to MSc. Tamara Figueredo, and Dra. Patricia González Díaz for their suggestions. We greatly thank our editor (Juan Pablo Quimbayo) and reviewers for their valuable comments and suggestions on our manuscript.

### Funding
The authors received no funding for this work.

### Competing Interests
The authors declare there are no competing interests.

## Author Contributions

- Andy Joel Corso performed the experiments, analyzed the data, prepared figures and/or tables, authored or reviewed drafts of the article, and approved the final draft.
- Fabián Pina-Amargós conceived and designed the experiments, analyzed the data, authored or reviewed drafts of the article, and approved the final draft.
- Leandro Rodriguez-Viera conceived and designed the experiments, performed the experiments, analyzed the data, prepared figures and/or tables, authored or reviewed drafts of the article, and approved the final draft.

## Field Study Permissions

The following information was supplied relating to field study approvals (i.e., approving body and any reference numbers):

Ministerio de Ciencia , tecnologia y mediao Ambiente. Permiso # 5_

## Data Availability

The raw measurements are available in the Supplementary Files. The raw data shows all the number of cleaning stations and the abundance and number of species of clients and cleaners associated with them. Additionally, interaction networks were provide for the three regions and the entire archipielago based on a total of 150 minutes' video records of active cleaning stations

## Supplemental Information

Supplemental information for this article can be found online at http://dx.doi.org/10.7717/peerj.16524#supplemental-information.

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
