# Peer review of "Cleaning symbiosis in coral reefs of Jardines de la Reina National Park"

_PeerJ, doi:10.7717/peerj.16524_

## Round 0.1 · original submission · Major Revisions

Dear authors,

I would like to say sorry to you for the long time waiting to send the decision on your manuscript. Your paper was reviewed for two different researchers and both have opinions very different about your manuscript, thus I requested a third reviewer before I take my decision. I believe that your paper is very important to region however you need to review all the comments that were mentioned for the reviewer 1 who reject initially your manuscript.

All the best

Juan Pablo

Reviewer 1 ·

Basic reporting

This study poses interesting questions, but the question regarding protection gradients is currently lost in the manuscript. When the authors discuss a protection gradient, I assumed that they would be analysing cleaning patterns over a continuous/ordinal variable (e.g. high, medium, low protection). Instead the authors discuss their findings in relation to central, eastern and western regions - this makes it difficult for the reader to pull out the main findings. It would therefore help the reader if the authors explained their findings in relation to the protection gradient, rather than comparing central region with east region etc (e.g. lines 234 - 238). Overall, the authors should consider significantly condensing the results and discussion sections to increase the clarity of their manuscript. For example, listing top cleaners and clients is unnecessary as they are presented in both Figure 2 and Table 2/3. The authors should also consider explaining further what their results mean in the results section itself. For example, it would help the reader if the authors included a reminder statement clarifying what functional redundancy means in this context (lines 251 - 253).

Sadly, I found this study very difficult to read and follow, it was therefore difficult to extract what the authors had done, and what the results showed. The clarity of the manuscript would be increased if the authors improved the English language used in text. For example, the manuscript is comprised of many long sentences (especially in the Discussion) which are difficult to follow (e.g. lines 86 – 90, 263 – 267). Furthermore, the authors should ensure that the English used is clear (e.g. lines 366, 240) and avoids colloquial terms (e.g. line 290: "strongly confirmed", line 281, line 324: "lax and loose nature"). The manuscript would also benefit if fewer abbreviations were used (e.g. CR, ER, JRNP), and where used, were re-defined multiple times in text. Personally, however, I don't think abbreviations are necessary for this manuscript. The authors should also attempt to ensure that their sentences are specific, and clearly define what is being addressed (e.g. avoid “Although our study did not find significant differences in some variables studied” line 363 – 364). The authors should also take great care in explaining what their results mean without using complex words with no definition/explanation (e.g. what is meant by “to be more specialised and functionally complementary”).

I also found the majority of included figures not very informative. For example, while the map shows sampling areas it does not show the protection gradient that is being studied. The authors also present results figures as only mean points ± mse bars and no raw data points are shown. The authors should consider showing the raw data on these plots (e.g. values per site/transect), and grouping multiple figures (e.g. 3,4,5) together in panels rather than individual plots. It would also be more informative to a reader to replace region with MPA protection level e.g. high, medium, low etc as this relates more clearly to the questions posed by the authors.

Data is provided but it is not sufficient (please see below).

Experimental design

I am concerned that the statistics used (e.g. Kruskall-Wallis) are too simplistic for the sampling design. The authors sampled multiple sites (often with multiple, but uneven numbers of transects) and these were grouped together at a region level (despite there being different numbers of sites sampled within region, from Table 1). Surveys were also conducted in three different years (2019, 2021 and 2022). The authors, I believe, did remove some sites so that the same number of sampling units were used per region, but it is not clear how/why the authors decided to remove one site/transect over another. It is also not clear how n = 43 has been derived, as that is not a multiple of three (e.g. west, east, central). The authors should consider using a generalised linear mixed model to analyse their data. This would take into account, transects, sites, year, protection levels and provide a more comprehensive and robust analysis. For example the authors could specify protection level and year etc as a main effect, and site as a random term. It would also be beneficial not to remove data points (e.g. use all 96 transects). Regarding the networks, the authors would also need to account for the uneven sampling as this will influence interaction abundances etc (e.g. greater sampling effort = more likely to observe an interaction). It may be beneficial if the authors create networks for each transect rather than region if it is not possible to account for uneven sampling effort. The authors could then use network metrics as individual datapoints within regions to ask how network structure changed with protection gradient.

Validity of the findings

Overall, it is sadly difficult to assess whether results are valid (please see statistics comments above). The authors have, however discussed their results in great detail, and have clearly thought about their meaning. It would improve clarity, however if the authors provided a paragraph at the start of the discussion stating how their results linked to the questions posed in the introduction. That would make it easier for the reader to glean key results. The discussion could also be condensed to increase clarity - there is a lot of information provided, and detracts from the main findings. For example, the authors should consider combining points on cleaning symbiosis across a protection gradient and network indices as they are complementary. The authors should also avoid talking about results that were non-significant as though they were significant. For example, the authors often discuss the tendency of results to follow a particular pattern, when statistically p values are far from significant (e.g. p = 0.5709, p = 0.9679) and so there is no evidence for the result being discussed (e.g lines 363 - 365, current figures and p values do not suggest a clear tendency for values to be higher in central region, especially as data points used in analyses are not provided on figures). The authors should instead discuss, in an ecological context, why they got the significant/non-significant results they did (following a re-analysis).

The data provided is not sufficient - the authors should consider presenting species counts/cleaning interaction count data at a transect level within each site in the same data file. It is currently not clear what the data files are showing and how they were used in analyses. For example, on "peerj-81505-Supplementary_file_1" it is not clear what the "Cleaning station", "Client" and "Cleaner" variables represent, especially when the same cleaner species (e.g. Elacatinus sp.) is listed multiple times.

Reviewer 2 ·

Basic reporting

No comment

Experimental design

It is not clear to me how long you observe each cleaning station. The authors mentioned that they did transect surveys but when a cleaning station was found, did you stop the transects and start observing the fish at the cleaning stations? Can you give more information about the time you spent in each cleaning station, please?

Validity of the findings

No comment

Additional comments

Overview
In this manuscript, the authors investigated fish cleaning interactions at Jardines de la Reina National Park, Cuba, which is subjected to different levels of protection. To do so, they did transect surveys on various reefs in the National Park (that differ in terms of protection level), with a special focus on cleaning stations from 6 different cleaners and client species diversity and abundance. The authors did not find significant differences between the regions, but the central region did in fact, show the highest values of cleaning stations density, client and cleaner abundance, and also exhibit the highest specialisation. Interestingly, this central region has indeed a higher level of protection. The manuscript is well-written, and I only have minor comments. A few sentences could be a bit clearer in the methods section but other than that, I did not find major issues. Therefore, I think that after taking into consideration the comments below, this manuscript will be suited for publication in PeerJ.

Introduction
L 57: missing reference at the end of the sentence.
L 75: Not sure about this statement here. The reference that the authors mention is limited to the terrestrial realm but they mentioned the Caribbean Sea. Please, substitute the reference with a more accurate one.

Methods
Sampling methodology
It is not clear to me for how long you observe each cleaning station. The authors mentioned that they did transect surveys but when a cleaning station was found, did you stop the transects and start observing the fish at the cleaning stations? Can you give more information about the time you spent in each cleaning station, please?
L 136: “Each bite by the cleaner on the client’s body was assumed to be an interaction”. With some of the species of cleaners that you recorded here in your study, such as the shrimps and Elacatinus sp., it is really hard (almost impossible) to distinguish clear bites. I recommend you specify for which species you did consider the bites as an interaction or remove this sentence. Interaction is where there is clear communication between the cleaner and the client and following this, there is skin-to-skin contact between them; therefore, that’s enough to be considered as cleaning interaction.

Results
L 186 – 187: Can you mention here which one are fish and which one are shrimps for unfamiliar readers with these families. Something like: “Six species, belonging to two families of fish: Labridae (2 species), Gobiidae (2 species); and two families of shrimps: Palaemonidae (1 species), and Stenopodidae (1 species) …”
L 189: I will suggest calling them Elacatinus sp. instead of mentioning both species and the word “complex” every time.
L 203 – 210: I suggest adding the family of the client fish species you mention in this paragraph for readers that are not familiar with species names.
L 228: There are two times the words “cleaning stations”. Remove the second time it’s mentioned.

Discussion
L 290: Remove the “,” after Sikkel.
L 301-306: This sentence is really long, please rewrite it for clarification.

Figures
Could you group Fig. 3 from 7 in one Figure panel maybe? Or maybe have the clients' figures on 1 panel and the cleaners' figures on another one. It just looks like a lot of figures to me to include in the paper.

Figure 2:
It would be good to have the entire name of the client species on the right side.
Also, I suggest grouping them by family (adding the family name) for easier interpretation. Finally, can you please add the species name in italic?

Table 1:
I think this table could go to the supplemental files. This information is great to have but will fit better in the supp. Files.

Reviewer 3 ·

Basic reporting

First, I congratulate the authors of this study. The manuscript is well written and I believe it could contribute to the field. This is an original work and fits within the journal's scope. However, different improvements could be made to reach the necessary quality for publication. Methods and results have different issues that must be addressed prior to acceptance. References used need some improvements as I believe there are some important works published in the literature that could contribute to the overall construction and discussion in this manuscript. Please, read the specific comments in the .pdf file.

Experimental design

The methods are, in general, well written and describe correctly and in detail the sampling design and analytical approaches.

The 'functional group', however, requires some attention and certainly reclassification. This change may not alter substantially the results but are necessary for avoiding misinformation in the published material. As some results and conclusions are based on the 'functional' categories, this issue must be addressed prior to publication.

Validity of the findings

No comment.

Annotated reviews are not available for download in order to protect the identity of reviewers who chose to remain anonymous.

---

## Round 0.2 · Minor Revisions

Dear Authors,

Having thoroughly reviewed your manuscript, I concur with the reviewer's observation that significant differences among the various levels of marine protection were not identified. Your discussion appears to revolve around these findings, which might inadvertently lead to a misinterpretation of your results.

With this in mind, I strongly recommend that you address all the concerns raised by the reviewer and prepare a revised version of the manuscript. I believe that this revised version could make a substantial contribution to our understanding of the intricate ecological interactions within the Caribbean.

Best regards,

Juan Pablo

Reviewer 2 ·

Basic reporting

The authors took into consideration the comments and now is ready for publication in my opinion. Congratulations!

Experimental design

no comment

Validity of the findings

no comment

Reviewer 3 ·

Basic reporting

The manuscript has been substantially improved since the last version and I congratulate the authors for attending to the previously addressed problems. The manuscript is clearer now, but other problems still need to be solved. For example, the approach in discussing the comparisons among the different areas with different protection status is problematic. There was no difference found, so there is no reason in trying to impose possible trends with numbers there were not statistically different.

Experimental design

Detecting and quantifying the effect of protection status in MPAs requires an understanding of spatiotemporal variability in the system and the type of comparison used, among other factors such as fishing pressure and species traits. Usually, finding true replicates of protected reefs is more challenging, but this problem does not justify making inferences in the absence of control (no protection status) sites. The comparison performed here may be problematic while trying to establish the causal effect of protection on the cleaning behaviour / community structure of the studied areas, as the experimental design did not randomly select treatment (protected site) and control (unprotected site) sites.

Inferring causality and determining whether variation in the cleaning stations and structure can be attributed to the different level of protection within protected areas, it is important to investigate heterogeneity of confounding factors or covariates over time and space. The authors recognise that their sampling methods added biases and limitations to their data and analysis, and, therefore, their results. Different sampling designs could partially control for confounding variables while accounting for pre-existing natural differences and distinguish the effect of environmental interventions from other sources of spatiotemporal variation. The effect of several covariates, such as different habitat characteristics between sites that likely affected the fish community and their behaviour were not taken into account (although the authors cite a work that could rule it out). Finally, it is important to highlight that disregarding confounding covariates might overestimate the effects of protected areas.

Validity of the findings

In the discussion, the most outstanding concern here is the fact that although the authors recognise that they found no difference in cleaning behaviour (and other metrics) among the different regions or status of protection in the JRNP, they seem to insist in pose differences among them. Instead of discussing why no differences were found, the authors insist in showing different values for the analysed metrics that were not statistically different. This approach might deviate the readers from the main finding (the areas do not differ among the analysed metrics) and confound them on the importance and effectiveness of the different levels of protection. I suggest the authors to give a careful look at the section "Effect of protection gradient on cleaning symbiosis" in the Discussion and rethink how they approach their findings.

Additional comments

I congratulate the authors for the improvement made throughout the manuscript since the first submitted version. The questions of the work are clearer now, the functional grouping of the fishes was corrected as well as the many issues pointed before, including the quality and presentation of the figures. However, as pointed above, some other issues must be addressed. See comments above and in the manuscript.

Annotated reviews are not available for download in order to protect the identity of reviewers who chose to remain anonymous.

---

## Round 0.3 · Minor Revisions

Dear Authors,

Thank you for submitting the revised version of your manuscript. Before accepting this paper, I would like to review some details that were addressed in the Rebuttal letter but have not been incorporated into the manuscript.

Rebuttal letter:
Authors "We also ran some generalized mixed linear models (GLMM) analyses taking sites as a random factor in the previous revision submission, and most of them resulted in sites not being a main driver of diversity. We think that this way, we controlled for an important part of the environmental variability among the sites. We have added this information in the methods section".

However in the text:

To assess the existence of a possible effect of the protection gradient on symbiotic cleaning interactions, MPA regions were compared according to number of cleaning stations, diversity of cleaner species, diversity of client species, total abundance of cleaners and total abundance of clients. This was done by using a smaller sample size (N = 37) than the initial one, so that three
regions had an approximately similar number of sample units. Variables were tested for homoscedasticity (Levene test) and to verify if they fitted to a normal distribution (Kolmogorov-Smirnov test). Non-parametric Kuskal-Wallis test was used to verify statistical differences (α =146 0.05) among regions


On the other hand,

In my opinion, a Non-parametric Kuskal-Wallis test is not the best analysis considering that several GLM, GAM, GLMM, and GAMM consider different types of distributions. These analyses are more robust than a Kuskal-Wallis test. I suggest applying some other model.

Please check this point and review the manuscript.

All the best

Juan Pablo

---

## Round 0.4 · accepted · Accept

Dear authors,

I am pleased to communicate that your paper was accepted. Thanks for considering this journal.

All the best

Juan Pablo